# A shared numerical representation for action and perception

Giovanni Anobile[1,2†], Roberto Arrighi[1†], Irene Togoli[1], David Charles Burr[1,3,4*]

[1]Department of Neuroscience, Psychology, Pharmacology and Child Health, University of Florence, Florence, Italy; [2]Department of Developmental Neuroscience, Stella Maris Scientific Institute, Pisa, Italy; [3]Institute of Neuroscience, National Research Council, Pisa, Italy; [4]School of Psychology, University of Western Australia, Perth, Australia

**Abstract** Humans and other species have perceptual mechanisms dedicated to estimating approximate quantity: a *sense of number*. Here we show a clear interaction between self-produced actions and the perceived numerosity of subsequent visual stimuli. A short period of rapid finger-tapping (without sensory feedback) caused subjects to underestimate the number of visual stimuli presented near the tapping region; and a period of slow tapping caused overestimation. The distortions occurred both for stimuli presented sequentially (series of flashes) and simultaneously (clouds of dots); both for magnitude estimation and forced-choice comparison. The adaptation was spatially selective, primarily in external, real-world coordinates. Our results sit well with studies reporting links between perception and action, showing that vision and action share mechanisms that encode numbers: a generalized *number sense*, which estimates the number of self-generated as well as external events.

**\*For correspondence:** dave@in.cnr.it

[†]These authors contributed equally to this work

**Competing interests:** The authors declare that no competing interests exist.

## Introduction

Animals, including humans, estimate spontaneously and reasonably accurately the approximate quantity of arrays of objects, without recourse to other forms of representation, such as density (*Cicchini et al., 2016*). Even newborn infants of less than 3 days show selective habituation to number (*Izard et al., 2009*). There is now very good evidence in both human and non-human primates that number is encoded by intraparietal and prefrontal cortex (*Castelli et al., 2006*; *Dehaene et al., 2003*; *Harvey et al., 2013*; *Nieder, 2005*, *2012*, *2016*; *Nieder et al., 2006*; *Nieder and Miller, 2004*; *Piazza and Eger, 2016*; *Piazza et al., 2004*, *2007*), even in numerically naive monkeys (*Viswanathan and Nieder, 2013*). All these studies point to the existence of a *visual sense of number* within a parietal–frontal network (*Dehaene, 2011*).

A truly abstract sense of number should be capable of encoding the numerosity of any set of discrete elements, displayed simultaneously or sequentially, in whatever sensory modality. Some evidence exists for such a generalized number sense. Neurons in the lateral prefrontal cortex (lPFC) of behaving monkeys encode numerosity for both auditory and visual sensory modalities, suggesting supra-modal numerosity processing (*Nieder, 2012*). Another study reported separate populations of neurons in the intraparietal sulcus (IPS) responding selectively to sequential or simultaneous numerical displays, while a third set of neurons showed numerosity selectivity for both simultaneous and sequential presentations, suggesting that the information about spatial and temporal numerosity converges to a more abstract representation (*Nieder et al., 2006*). There is also evidence from functional imaging in humans for a right lateralized fronto-parietal circuit activated by both auditory and visual number sequences, and that right IPS is involved in processing both sequential and simultaneous numerosity formats (*Castelli et al., 2006*; *Piazza et al., 2006*).

**eLife digest** Humans and many other animals have the ability to make spontaneous and rapid estimates of the approximate number of items that they can see. This sense of number, or "numbersense", is particularly important in humans, as evidence suggests that it lays the groundwork for acquiring mathematical skills.

Researchers have many questions about numbersense. Is it a kind of perception? Or does it require more active thought, like counting? Do people have the same sense of number when they view, hear or touch items that depict the same number? Having a sense of number is essential for carrying out certain actions, like the following the steps in a dance, but the connection between action and numbersense is not entirely clear.

A process called adaptation means that viewing specific stimuli for a period of time can affect what people think they see subsequently. For example, viewing large numbers of dots makes subsequent smaller groups of dots seem like they contain fewer dots than they actually do. Anobile, Arrighi et al. have now investigated the link between action and numbersense by asking volunteers to tap one hand either rapidly or slowly in one spot for a short time. The volunteers were then shown a series flashes or a cloud of dots in the region where they had been tapping and asked to estimate the number of flashes or dots.

After fast tapping, the volunteers greatly underestimated the numbers of flashes or dots that they saw; after slow tapping, they overestimated the numbers. However, if the images were shown far away from where the volunteers had been tapping, their estimates were more accurate.

Overall, the results suggest that adaptation is controlled by space-specific sensory mechanisms rather than some kind of active counting. Furthermore, numbersense appears to have a generalized form that is shared by the brain regions responsible for perception and action. Because numbersense and mathematical ability are linked, this strong connection between action and number perception may have important implications for understanding and treating math-related learning disabilities. Anobile, Arrighi et al. next plan to study how movement-driven adaptation affects numbersense in children and adults with these conditions.

Psychophysical evidence showing little cost in cross-modal or cross-format matching also points to a common number sense spanning sensory modalities and formats. For example, human adults are very efficient in making cross-modal and cross-format judgments, with very little cost in either accuracy or reaction times when comparing auditory with visual temporal sequences or dot arrays (*Barth et al., 2003*; *Brannon, 2003*). Developmental work also show similar accuracy in pre-school-ers for comparing spatial array of dots either with other spatial arrays, or with sequences of sounds (*Barth et al., 2005*). Preferential-looking studies show infants prefer to look at screens displaying adults faces numerically matched with the soundtrack of adult voices (*Jordan and Brannon, 2006*), and abstract visual ensembles (shapes) numerically matched with on-going sequence of sounds (*Izard et al., 2009*). However, not all agree: Tokita & Ishiguchi (*Tokita and Ishiguchi, 2012*) reported significantly lower precision for cross-format number comparisons in adults, than for within format comparisons.

That there is little or no cost in these matches is certainly indicative of efficient transfer of information between senses, but says little about the mechanisms involved. The match is made at the decision level, so the interaction could be at any stage up to and including decision mechanisms. One of the more powerful psychophysical techniques to probe mechanisms is adaptation (*Mollon, 1974*; *Thompson and Burr, 2009*). Number, like most other primary visual attributes, is also highly suscep-tible to adaptation (*Schwiedrzik et al., 2016*; *Burr and Ross, 2008*): visually inspecting for a few seconds a large number of items results in the perceived numerosity of a subsequent ensemble to be strongly underestimated, and vice-versa after adaptation to low numbers (*Burr and Ross, 2008*). More recently we have shown that adaptation to numerosity also occurs with sequentially presented stimuli, and that the adaptation effects are both cross-modal and cross-format (*Arrighi et al., 2014*): adapting to sequences of tones affects the perceived numerosity of a subsequently presented series of flashes (and *vice versa*), and adapting to sequences of flashes affects the perceived numerosity of

spatial arrays of items. Importantly, the adaptation was spatially selective, in external rather than eye-centered coordinates, suggesting it has a perceptual rather than cognitive basis. fMRI studies have demonstrated BOLD activity selective for adaptation to numerosity in the human parietal sulcus (*Piazza et al., 2004*, *2007*; *Castaldi et al., 2014*; *He et al., 2015*).

All these studies strongly suggest that the number sense is a high-generalized system, capable of combining numerical information from different senses, and across different formats. Numerical information is also relevant for the production of specific action sequences, from dance routines to more simple repetitive behavioral tasks. A few studies point to an interconnection between numerosity and motor control. For example, neurons in area 5 of the superior parietal lobule of monkey show a clear selectivity for the number of self-produced actions, and inactivation of the area impedes number-based tasks (*Sawamura et al., 2002*, *2010*). Other work has shown that the left ventral premotor cortex is activated by counting successive sensory stimuli (*Kansaku et al., 2007*), and that the human cerebellum shows strong activation for simple numerical calculations (*Arsalidou and Taylor, 2011*).

The existence of anatomical and functional connections between number and action-generation systems raise the possibility that number-for-action could be encoded within a truly abstract numerosity mechanism. To test this idea, we measured cross-adaptation between motor repetitions and perception of numerosity. The results show that adapting to self-generated action does affect the representations of numerosity of external events, both sequential (series of flashes) and simultaneous (dots ensembles), and that the adaptation is spatially selective in external, not hand-centered coordinates.

## Results

Each trial began with a motor adaptation phase in which participants performed tapping movements for six seconds, under two different conditions (tested on separate sessions): 'high adaptation', where participants were asked to tap as quickly as possible (average 5–6 taps/s); and 'low adaptation' where they tapped more slowly (average 1.12 taps/s: see Materials and methods and Figure 5). After the adaptation phase, the test stimulus – either a sequence of flashes or a cloud of dots (tested on separate sessions) – was randomly displayed either to the same side of the screen where the hand had been tapping, or to the symmetrically opposite side. Participants estimated the numerosity of the test stimulus, which varied randomly from trial to trial within the range 6–14. To minimize sensory feedback, participants were placed in a dark room and wore soundproof headphones, and tapped in mid-air behind the computer screen without touching any surface.

The results are shown in *Figure 1*. Panels A & B show numerosity estimates averaged over all subjects as a function of the physical numerosities displayed. When the test stimulus was displayed on the right side of the screen (where the adaptation had occurred), rapid tapping caused a consistent underestimation of the numerosity of the test, while slow tapping caused an overestimation. The adaptation effects were similarly strong for when the test was a sequence of flashes (*Figure 1A*) as when it was an array of dots presented simultaneously (*Figure 1B*). Interestingly, the effect occurred only when the stimuli were presented on the same side as the tapping hand (the right side): when presented on the other (left) side, adaptation produced no consistent effect (*Figure 1A and B* open symbols).

We defined *adaptation magnitude* as the percentage difference in perceived numerosity after adaptation to fast or slow tapping, averaged over all numerosities. For sequential and simultaneous presentations, the adaptation magnitude averaged across subjects (filled symbols in *Figure 1C and D*) was around 20 and 25% respectively for stimuli presented to the adapted location, a very strong effect. For stimuli presented to the unadapted location, the average effect was only 4 & 2%.

We also calculated adaptation magnitude for individual subjects. *Figure 1C and D* plot adaptation magnitudes for the congruent condition (where the visual stimuli were presented to the right side), against the incongruent condition (stimuli to the left side). All subjects showed a significant effect in the congruent condition (error bars 1 sem), but very little effect in the incongruent condition. ANOVA showed that the congruent conditions were highly significant ($F_{(1,32)}$ = 70.219, p = 0.001, $\eta^2$ = 0.29, Cohen's d = 1.278 and $F_{(1,48)}$ = 47.176, p = 0.0004, $\eta^2$ = 0.217, Cohen's d = 1.062 for sequential and simultaneous condition respectively), while the non-congruent conditions

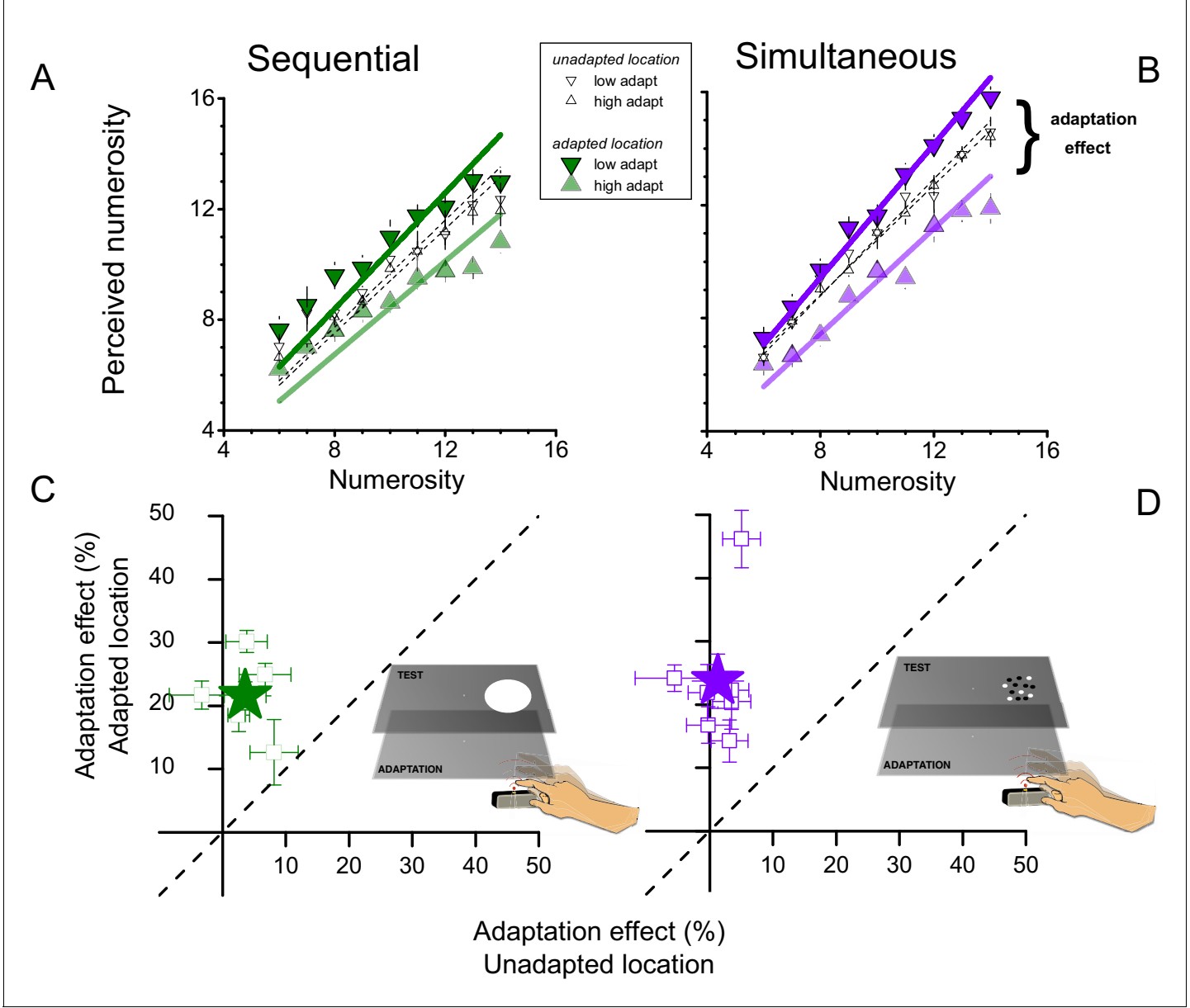

**Figure 1.** Effects of motor adaptation on perceived numerosity. (**A** and **B**) Average perceived numerosity as a function of physical numerosity for slow tapping (downward triangles) and fast tapping (upward triangles), for sequential (left) and simultaneous (right) formats. Filled symbols indicate the conditions in which stimuli were spatially congruent with the tapping region, small open symbols to estimates obtained for the unadapted location (left-hand side). (**C** and **D**). Adaptation magnitudes for individual subjects when test and tapping were spatially congruent, plotted against the spatially incongruent condition. Stars reports averages, squares single subject data. Error bars refer to ± 1 SEM.

were weak and insignificant ($\approx$ 4% effect, $F_{(1,32)} = 1.403$, $p = 0.302$, $\eta^2 = 0.007$, Cohen's d = 0.167 and $\approx$ 2% effect, $F_{(1,48)} = 0.919$, $p = 0.375$, $\eta^2 = 0.008$, Cohen's d = 0.179). That the adaptation is spatially specific suggests it is of a perceptual rather than cognitive nature, and unlikely to result from a response bias or any other generalized artifact.

This first experiment revealed two clear results: that motor adaptation affects visual estimates of numerosity, for both sequential and simultaneous displays; and that the adaptation is spatially specific. The spatial specificity suggests that the effect is not a high-level, cognitive phenomenon (such as 'internal counting'), but perceptual in nature, mediated by neural mechanisms with circumscribed receptive fields. To verify the robustness of the spatial selectivity, and to understand it better, we

repeated the experiment with a new subject pool, changing the tapping hand and location. In this experiment we tested only the simultaneous presentation, as this is the most revealing (and surprising) result.

The violet symbols of *Figure 2A* replicate the results of the previous experiment, tapping with the right (dominant) hand and testing on both right and left sides (randomly interleaved): the adaptation effect was again strong for stimuli presented on the same side (filled symbols), and non-existent for stimuli on the other side (open symbols) ($F_{(1,40)}$ = 70.207, p = 0.000397, $\eta^2$ = 0.116, Cohen's d = 0.724; $F_{(1,40)}$ = 2.036, p=0.213, $\eta^2$ = 0.0019, Cohen's d = 0.0873; for adapted and unadapted location respectively). The red symbols of *Figure 2B* show the results for tapping on the left with the

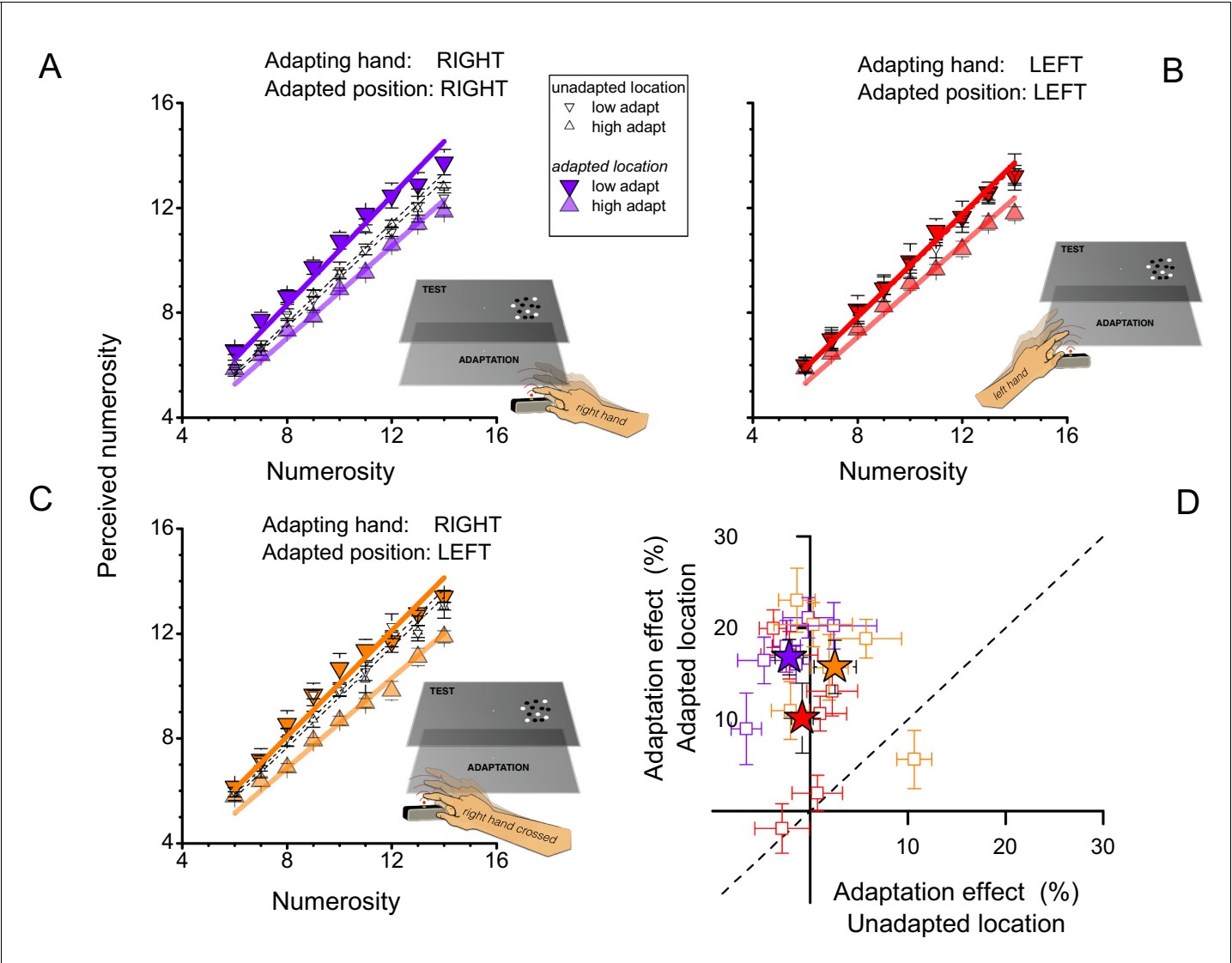

**Figure 2.** Reference frame of motor adaptation. (A) Average perceived numerosity as a function of physical numerosity for the slow-and fast-tapping conditions (downward and upward triangles respectively), for right-hand tapping. Filled symbols refer to trials when the stimuli were presented in the spatial region where the subjects had tapped (right side) small open symbols to trials when the stimuli were presented on the other side. This data replicates *Figure 1B* with a fresh subject pool. (B) Same as A, except subjects tapped with their left hands. Filled symbols refer to testing in the same spatial region where the subjects had tapped (left side), small open symbols to the right side. Other conventions like A. (C) Same as A, except the right hand tapped on the left side of the screen. Filled symbols refer to testing on the same spatial region where the subjects had tapped (left side), small open symbols to the right side. (D) Adaptation magnitudes for individual subjects when test and tapping were spatially congruent, plotted against the spatially incongruent condition. Color-coding as for A, B and C (purple: right hand, right side; red: left hand, left side; orange: right hand, left side). Stars reports averages, squares single subject data. Error bars refer to ± 1 SEM.

left (non-dominant) hand: again the effects occurred only for visual stimuli presented on the congruent side (left), although they were somewhat weaker ($F_{(1,40)}$ = 9.305, p = 0.028, $\eta^2$ = 0.05, Cohen's d = 0.4588; $F_{(1,40)}$ = 0.265, p = 0.629, $\eta^2$ = 0.001, Cohen's d = 0.0633; for adapted and unadapted location respectively). *Figure 2C* shows results for tapping with the dominant (right) hand on the left side of the screen. Here, adaptation was found only for stimuli presented to the left side of the screen, suggesting that it is spatially selective in external rather than hand-centered coordinates ($F_{(1,40)}$ = 36.840, p = 0.002, $\eta^2$ = 0.104, Cohen's d = 0.6814; $F_{(1,40)}$ = 1.380, p = 0.293, $\eta^2$ = 0.0023, Cohen's d = 0.096; for adapted and unadapted location respectively). *Figure 2D* shows the results for all six subjects. There is some variability between subjects, particular in the crossed condition, where one subject showed adaptation to stimuli on the right after tapping on the left with the right hand, but by and large the individual data reinforce the group data.

In the previous experiment, subjects tapped in mid air to minimize sensory feedback. In the next series of experiments we manipulated the amount of sensory feedback in the adaptation phase to examine interactions between sensory and motor signals. In the first condition (tactile only), subjects tapped a mouse behind the monitor, allowing for tactile feedback (*Figure 3A*). The adaptation effect in this condition was strong, around 20% ($F_{(1,40)}$ = 743.738, p = 0.0001, $\eta^2$ = 0.203, Cohen's d = 1.009). In the next condition (visual and tactile), the monitor accompanied each mouse-tap with a flash, to give visual as well as tactile feedback. Despite the extra feedback, adaptation remained around 20%, ($F_{(1,40)}$ = 36.746, p = 0.002, $\eta^2$ = 0.184, Cohen's d = 0.949) as shown in panel B of *Figure 3*. The last adaptation condition (visual only) comprised a sequence of visual flashes whose rates were determined by the adapting motor routine of the previous conditions (visual and tactile). Again, the adaptation effect was found to be strong ($F_{(1,40)}$ = 61.740, p = 0.001, $\eta^2$ = 0.230, Cohen's d = 1.093), and similar to the other conditions, around 20% (*Figure 3C*), making these three adaptation conditions equally effective as tapping in mid-air ($F_{(4,29)}$ = 0.475, p = 0.754, $\eta^2$ = 0.07, Cohen's d = 0.548: see *Figure 3D*).

We also verified the results with a two-alternative forced-choice technique. Subjects adapted to high or low tapping rates, as in the first experiment (no tactile or visual feedback), then two clouds of dots were simultaneously presented to the right (adapted) and left (unadapted) positions. The numerosity of each stimulus varied from trial to trial over the range 5–20, and subjects indicated which stimulus appeared more numerous. *Figure 4A* plots average responses as a function of the difference between the right and the left stimulus (normalized to the average of the two numerosities), to yield psychometric functions. The effect of adaptation was again clear: adapting to low tapping rates shifts the curve to the left (compared with baseline), consistent with an overestimation of the perceived numerosity ($t_{(5)}$ = 3.285, p = 0.021, Cohen's d = 1.101) and high tapping rates caused the opposite effect, even if weaker ($t_{(5)}$ = 1.237, p = 0.27, Cohen's d = 0.558). The differences in the points of subjective equality (PSEs, given by the 50% point of the curves) of the two adapting conditions again gives an index of magnitude of adaptation, around 15%. *Figure 4B* shows the PSEs for adaptation to the two conditions. Despite some variability amongst subjects the effects are quite robust and statistical significant as shown by a two-tailed paired t-test: $t_{(5)}$ = 3.56, p = 0.029, Cohen's d = 1.612. This experiment confirms the main results with a different technique, and also confirms the spatial selectivity of the adaptation: if adaptation was not spatially selective, it would work equally on the presentations to the left and right sides, annulling the effect.

## Discussion

This study shows that estimates of numerosity, both sequential and simultaneous, are strongly biased after adapting to repetitive finger tapping: rapid tapping decreases apparent numerosity, slow tapping increases it. The effect is spatially selective, primarily in external rather than hand-centered coordinates.

There has been a long-standing debate as to whether adaptation effects operate on numerosity per se, or via texture-density mechanisms (*Burr and Ross, 2008*; *Anobile et al., 2014*, *2015*, *2016*; *Bell et al., 2015*; *Dakin et al., 2011*; *Durgin, 1995*, *2008*; *Morgan et al., 2014*; *Ross and Burr, 2012*; *Ross, 2010*; *Tibber et al., 2012*, *2013*). A similar argument could be made here: that the adaptation was to temporal frequency, rather than to numerosity. As with spatial adaptation, there are many reasons to suggest that this is unlikely. However, the cross-format

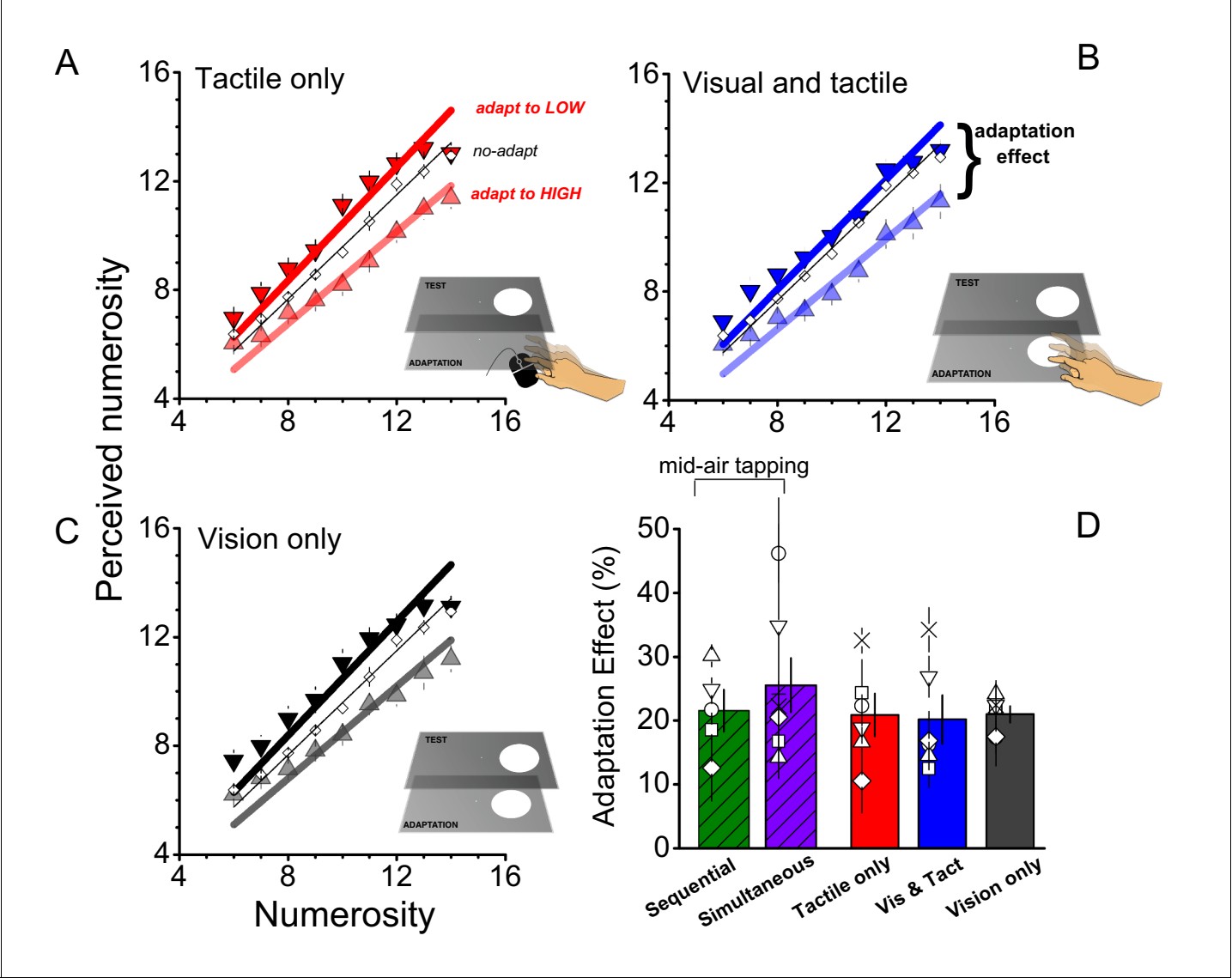

**Figure 3.** Role of sensory feedback of motor adaptation on perceived numerosity. (A), (B), (C) Average responses as a function of physical numerosity for slow adaptation (downward triangles), fast adaptation (upward triangles) and no adaptation (diamonds), for the three different conditions. (D) Bar graphs report the average adaptation effect for all adapting conditions (tactile only - red; visual and tactile - blue, visual only – black and the 2 conditions of Exp 1: sequential-green and simultaneous-violet). Open symbols show single subject data. Error bars report ± 1 SEM. All the conditions provided significant effects (all p-values < 0.05). The magnitude of the effect does not differ between conditions (p > 0.05).

adaptation (adapt to tapping sequence and test on dot array) clearly rules out this possibility: the spatial arrays are not temporally modulated. It is numerosity that is being adapted, not temporal frequency.

The current results reinforce the many previous studies (*Izard et al., 2009*; *Nieder, 2012*; *Nieder et al., 2006*; *Barth et al., 2003*; *Brannon, 2003*; *Barth et al., 2005*; *Jordan and Brannon, 2006*; *Arrighi et al., 2014*; *Jordan et al., 2005*) discussed in the introduction that point to the existence of a generalized sense of number. Most of these studies relied principally on cross-modal comparisons of number, which could occur at any processing stage, up to and including decision mechanisms. The spatial selectivity shown in our study suggests that the interaction is perceptual rather than cognitive: adapting on the left side did not affect stimuli on the right, and vice versa. Importantly, the specificity was in external coordinates, as adapting the left field with the right hand

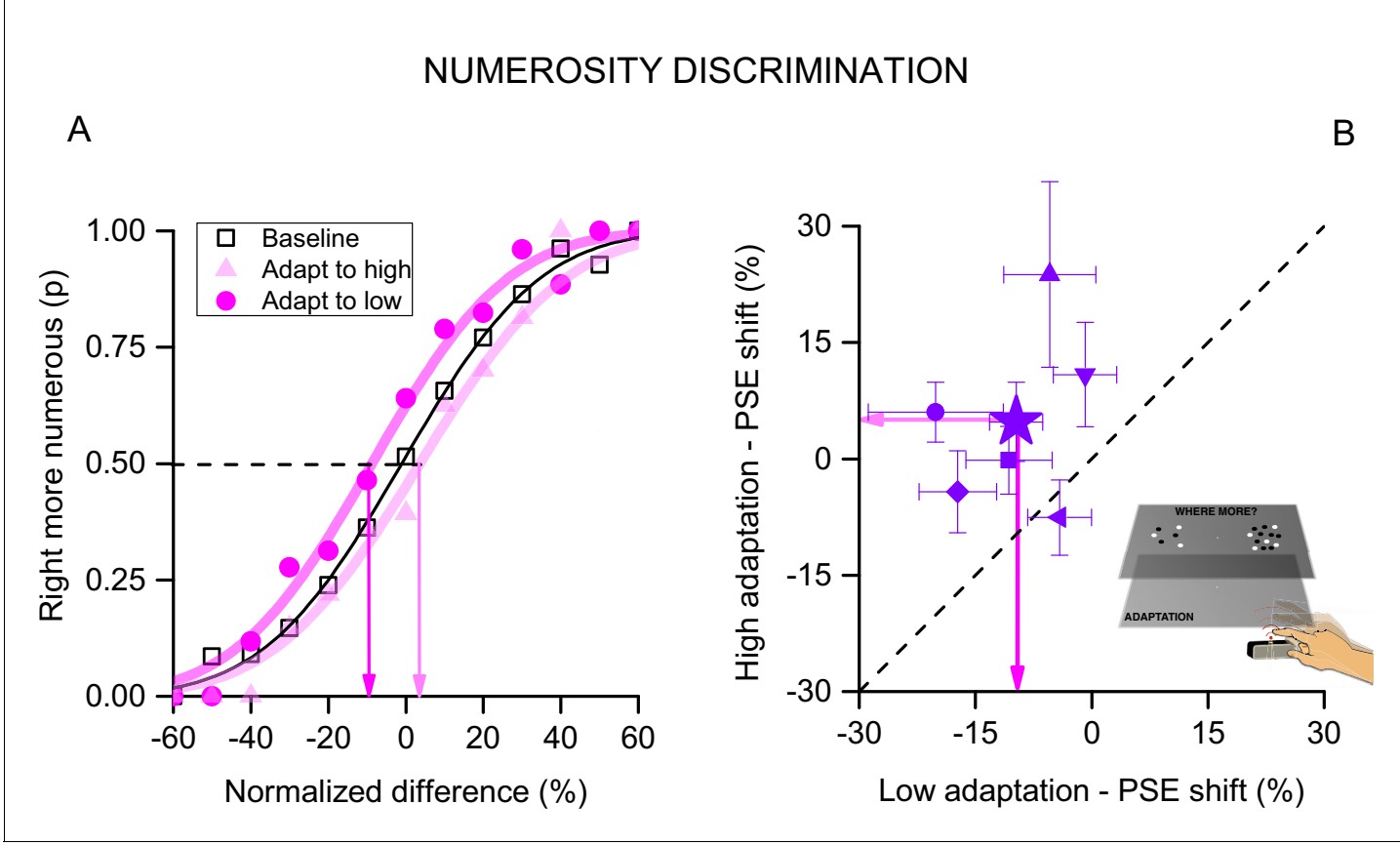

**Figure 4.** Forced-choice measurement of motor adaptation. (**A**) Psychophysical functions for pooled data (6 subjects) after adaptation to fast (light violet circles), slow (dark violet triangles) or no (black squares) tapping. The curves indicate the proportion of trials when the test (presented on the right, the same side of tapping) was seen as more numerous than the unadapted stimulus (presented on the left), as a function of the numerosity difference (normalized by the averaged of the two stimuli). Adaptation to slow tapping shifted the curve leftwards, showing that subjects were biased to perceive the stimulus as more numerous that it was; and adaptation to fast tapping shifted it rightwards. The point where the best-fitting curves pass 50% is considered the point of subjective equality (PSE, indicated by the coloured arrows). (**B**) PSEs for individual subjects after adaptation to fast tapping (ordinate) against those after adaptation to low motor repetitions (abscissa). The filled star shows results for data averaged across subjects. Error bars report ± 1 SEM.

caused adaptation for visual stimuli presented to the left, not the right visual field. This complements nicely the result of our previous study (*Arrighi et al., 2014*), where we showed that adaptation to visual sequences affects number perception of both sequential and simultaneous presentations, in a spatially selective manner. Interspersing an eye-movement between adaptation and test showed that the adaptation was spatially specific in external rather than eye-centered coordinates: as the current study shows the selectivity is external, not hand-centered. It would be interesting to look at the spatial tuning of the adaptation on a finer grain, to define the size of the adaptation field. The present study shows that the adaptation is at least broadly tuned, confined to a particular hemifield. It would be very informative to determine whether there was also selectivity within each hemifield, and on how fine a grain.

Some may find the spatial selectivity of the adaptation difficult to reconcile with the concept of a generalized, abstract sense of number. However, cross-modal effects can also show spatial selectivity. For example, cross-modal integration of visual and auditory (or tactile) information occurs only if the stimuli are spatially coincident (within certain bounds) (*Slutsky and Recanzone, 2001*). Similarly event time, which certainly transcends modalities, and also seems to be coded in parietal cortex (*Leon and Shadlen, 2003*), is affected by motion adaptation, in a spatially selective manner (*Burr et al., 2007*; *Fornaciai et al., 2016*). Interestingly, the spatial selectivity of the adaptation is in

external – not eye-based – coordinates as we observed for number, here and in the previous study (*Arrighi et al., 2014*).

We tested adaptation to action under various feedback conditions: visual and tactile, only visual, only tactile, and minimal feedback. All conditions produced similar amounts of adaptation. In the 'minimal feedback' conditions, where subjects tapped in mid-air, there was no tactile feedback from hitting a surface. We could not, however, remove all forms of kinaesthetic feedback, and therefore cannot be certain whether the adaptation signal was the intension to move, or the sensory proprioceptive feedback from the finger. But both are signals about action, whether they are 'inflow' or 'outflow'. It is interesting that this condition with reduced perceptual feedback produced the same amount of adaptation, as did the conditions with visual and/or tactile feedback. It is also interesting that the vision-only condition produced similar adaptation.

Many studies have suggested that vision and action are linked (*Arrighi et al., 2011*; *Goodale and Milner, 1992*). This study is a further clear example of their interconnection, in the encoding the numerosity of internally generated actions and externally generated events.

## Materials and methods

### Participants
A total of 15 adults (13 naïve to the purpose of the study, 2 author; mean age 27, all right-handed with normal or corrected-to-normal vision) participated in the numerosity estimation experiments. Six of them were tested in the sequential condition (test stimuli: sequences of flashes) and 7 of them in the 'simultaneous condition' in which test stimuli consisted of array of dots simultaneously presented. Three of these (2 author and 1 naïve subject) also participated, together with 3 additional naïve subjects (mean age of group: 28), in the second experiment investigating the reference frame of the motor adaptation after-effect. Eventually, six subjects (mean age of group: 28) were tested in the experiment concerning forced-choice discrimination of numerosity. All participants gave written informed consent. Experimental procedures were approved by the local ethics committee (Comitato Etico Pediatrico Regionale—Azienda Ospedaliero-Universitaria Meyer—Firenze FI) and are in line with the declaration of Helsinki.

### Stimuli
Stimuli were created and presented with Psychophysics toolbox for Matlab and displayed on a 60 Hz - 17", touch screen monitor (LG-FLATRON L1732P) placed at a subjects view distance of 57 cm. To eliminate auditory feedback, participants wore soundproof headphones. In some conditions, hand movements were monitored by an infrared motion sensor device (Leap motion controller - https://www.leapmotion.com/) running at 60 Hz.

### Statistical analyses
During the design of the experiments we computed an appropriate sample size to confidently report an effect of motion adaptation on perceived numerosity. Sample size was measured by means of a one-sample t-test assuming a value of 0 (no effect) as a Null Mean and retrieving a value for alternative mean and standard deviation from a previous study of our group (see Figures 4 and 5 in Arrighi, et al. [*Arrighi et al., 2014*]). The analysis revealed that with a sample size of 4, a power of 0.95 was achieved with an alpha level of 0.01. For this reason in all or experiments, we always tested a number of participants greater than 4 (see below for details).

We did not set any inclusion criteria for subject selection or their data: all data, for all experimental conditions, were analyzed and reported. In all conditions where subjects estimated numerosity we tested statistical significance with a $2 \times 9$ repeated measures ANOVA with test numerosity (9 levels for numerosity, range 6–14) and adaptation type (low and high) as main factors. Difference in the adaptation effects between the several adaptation conditions (visual, tactile, visual-tactile, and the two conditions with minimal feedback) were measured by a one-way ANOVA. In the numerosity discrimination task, difference in the adaptation effects for high and low adaptation were tested for statistical significance by mean of two-tailed paired t-test.

For t-test analyses we measured Cohen's d. For repeated measures ANOVA and regression analyses, we reported both Cohen's d and $\eta^2$. Here Cohen's d was measured transforming $\eta^2$ into Cohen's d (*Cohen, 1988*). All data are publicly available at Figshare (*Anobile et al., 2016*).

## Experimental procedure

### Numerosity estimation

During the adaptation phase, participants made a series of tapping movements on the left or right side of the screen until a white central fixation point turned red (the stop signal), and 1 s later the test stimulus was presented. Participants usually completed their current movement within 500 ms, so there was a 500 ms pause between movement-completion and test presentation. The program continuously monitored tapping in all conditions: if a tap occurred after the presentation of the test stimulus, the trial would be aborted: in practice this never occurred. For most experiments, subjects tapped with their dominant (right) hand on the right side of the screen. For the second study, however, we also tested tapping with the right hand on the left side, and with the left hand tapping on the left side.

Five separate adaptation conditions were tested. 1) 'Visual and tactile' (action with visual and tactile feedback): each tap on the monitor surface triggered the simultaneous appearance of a visual flash surrounding the zone where the finger touched the screen. 2) 'Only tactile': participants tapped on a mouse button located beneath the screen, without visual feedback. 3) 'Only visual': participants were presented with a sequence of visual events whose rate was taken from the previous the motor adaptation condition. 4 and 5) 'Minimal feedback': participants tapped beyond the screen without touching any surface, tapping with the hand floating between the screen and a infrared sensor device fixed on the desk. In one condition the test stimuli consisted of sequence of flashes (sequential) in the other test stimuli were ensembles of dot (simultaneous). The simultaneous condition was also used in the series of experiments shown in *Figure 2* in which we tested the reference frame of the motor adaptation effect. In one condition we replicated the previous paradigm (with fresh subjects) by asking subjects to tap with the dominant (right) hand on the right side of the screen. In another condition subjects tapped with the non-dominant (left) hand on the left side. In the third condition participants crossed their dominant (right) hand to tap on the left side of the screen.

Two adaptation levels were tested separately for each condition. In one we asked subjects to make as many taps as possible within the adaptation period (high adaption), in the other to tap at a far slower rate (low adaptation): see *Figure 5* for distributions of tapping rates. In all experiments, the adaptation phase lasted 6 s, and taps were always made with the right hand, on the right side of the monitor (hand placed 7 deg to the right of the central fixation point). After adaptation, the test phase started. In all conditions except 'simultaneous', test stimuli were a series of white disks (7 deg diameter), each presented for 40 ms within an interval of 2 s. To minimize temporal regularity, each disk was temporally jittered with the rule that two consecutive stimuli could not be displayed with an inter stimuli interval less then 40 ms (with a maximum ISI of 290 ms, in case of the lowest numerosity N = 6). In the simultaneous condition, test stimuli were circular clouds of dots (ensembles of half-white half-black non-overlapping dots, 0.3 deg diameter, presented for 250 ms within a circular region of 7 deg of diameter) centered at 7 deg eccentricity. In the two minimal feedback conditions (Sequential and Simultaneous), test stimuli were presented both in the adapted position and the opposite side (centred 7 deg to the left of the central fixation point), randomly selected trial-by-trial.

In all conditions, after presentation of the test stimuli a virtual numerical keypad was displayed for subjects to record their response by mouse-click. Nine test numerosities were used, 6–14 inclusive. Each participant performed about 260 trials (4/5 separate sessions), roughly equally divided between 'low' and 'high' adaptation and test numerosity levels (randomly selected trial-by-trial) leading to a total amount of trials of approximately 7500. The order of conditions was randomized between subjects.

Before starting testing, participants were familiarized with stimuli performing a block of 20 trials with sequential stimuli and 20 trials with dots stimuli. During the familiarization phase, we provided feedback of the exact number of items/events displayed. No motor (adaptation) training occurred during the training phase, and no feedback was provided during test phase. We defined an adaptation index (AI) as the average percentage change in perceived numerosity after high and low adaptation, averaged across all numerosity (*Equation 1*).

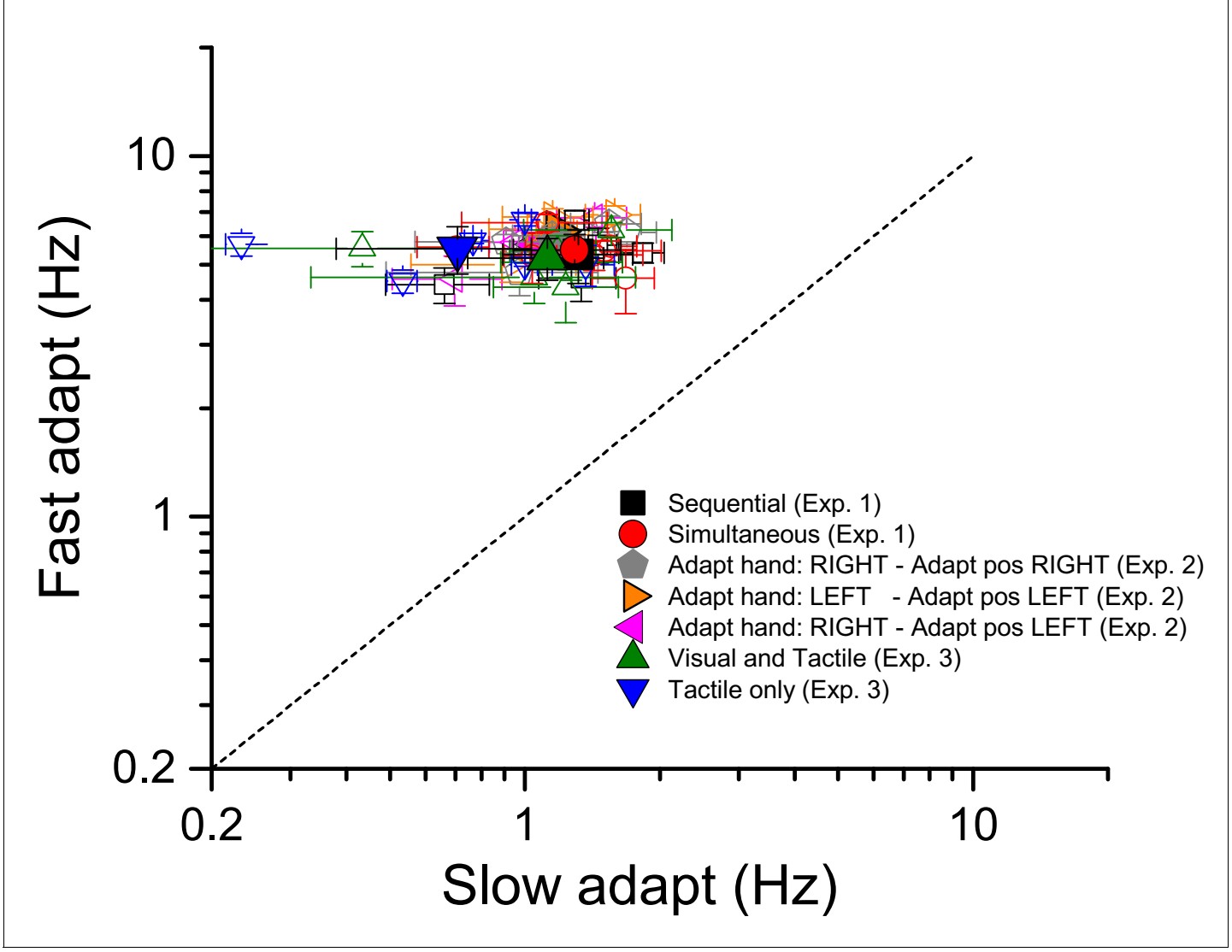

**Figure 5.** Tapping rates for high and low adaptation. Tapping rates (open symbols for single subject data; filled symbols for averages) for two different adaptation conditions: fast adaptation (ordinate) and slow adaptation (abscissa) for seven different experimental conditions. Black and red refers the two conditions in which subjects tapped in mid-air and then estimated numerosity of either sequential or simultaneous visual stimuli respectively. Gray, orange and violet refer to the three different versions of the simultaneous conditions devised to investigate the reference frame of adaptation: gray - subjects tapping with the right hand on the right side, orange - left hand on the left side and purple -right-hand on the left side. The adapting conditions in which subjects tapped on a surface (receiving tactile feedback) are indicated by green and blue symbols: green refers to the 'visual and tactile' condition in which participants tapped on touch-screen surface and were provided with visual feedback of their moving hand (visible) as well as by flashes on the monitor signaling the contact between the finger and the touch screen. Data in blue refer to the 'tactile only' condition in which subjects tapped on the mouse button placed beyond the screen (moving hand not visible).

$$\text{AI} = \left( \frac{100}{n} \sum_{j=1}^{n} \frac{\overline{RL}_J - \overline{RH}_J}{(\overline{RL}_J + \overline{RH}_J)/2} \right) \tag{1}$$

with $n = 9$, the number of numerosities tested (ranging from 6 up to 14), $\overline{RH}$ the averaged response to a given numerosity after high adaptation, $\overline{RL}$ the averaged response after low adaptation.

In the two-alternative forced-choice experiment subjects were simultaneously presented with two clouds of dots (like those described above) to the right and the left of the central fixation point, both centered at 7 deg. On each trial, the numerosity of the patch on the right hand side was

chosen at random between 5 and 20 dots; that on the left differed by a random value within the range ± 5 dots (capped between 5–20). Subjects were required to choose the more numerous. As there was variability in the numerosity of on both sides, subjects were not tempted to make a stereotypical response. In separate sessions numerosity discrimination was preceded by fast tapping, slow tapping or no-motor action (baseline). The effect of motor adaptation was measured as the difference in points of subjective equality (expressed as percentage) between high and low adaptation. For all experiments, tapping was always with the right hand.

## Tapping rates

*Figure 5* plots the tapping rate for the fast against the slow adaptation conditions, expressed as actions per second (Hz). Different colors and symbols refer to different experimental conditions (see caption). On average (across trials and conditions), when asked to tap quickly, participants tapped at a frequency of 5–6 Hz (for a total number of 30–36 tapping repetitions) with almost no difference between the adapting conditions: mean 5.33 ± 0.9; 5.48 ± 0.5; 5.2 ± 0.7; 5.54 ± 0.8; 6.19 ± 0.37; 5.69 ± 0.38; 5.67 ± 0.31 for the 'sequential', 'simultaneous', 'visual and tactile', 'tactile only', 'adapt with the right hand in the right space', 'adapt with the left hand in the left space' and 'adapt with the right hand in the left space' respectively. Also tapping frequencies for the condition in which subjects tapped slowly were similar across adapting conditions with all values ranging between 0.7 and 1.3 Hz (mean 1.31 ± 0.4; 1.29 ± 0.3; 1.12 ± 0.4; 0.7 ± 0.3; 1.18 ± 0.12; 1.07 ± 0.13; 1.18 ± 0.17 for the 'sequential', 'simultaneous', 'visual and tactile', 'tactile only', 'adapt with the right hand in the right space', 'adapt with the left hand in the left space' and 'adapt with the right hand in the left space' respectively). These data clearly indicate that regardless the tapping routine to be performed on a rigid surface or in mid-air, the tapping temporal dynamics were always very similar.

We also tested whether there was a correlation between faster tapping rate and adaptation effects. There was a slight, but non-significant tendency for faster tapping rates to be associated with lower adaptation. But as the correlation was not significant, we assume that variable tapping rates was not a cause for concern for the results of these experiments.

## Acknowledgements

This research was funded by the Italian Ministry of University and Research under the project 'Futuro in Ricerca' Grant number RBFR1332DJ, by the European Research Council under the Seventh Framework Programme (FPT/ 2007-2013, Early Sensory Cortex Plasticity and Adaptability in Human Adults) Grant number 338866 and from Italian Ministry of Health and by Tuscany Region under the project 'Ricerca Finalizzata', Grant n. GR-2013-02358262 to GA.

## Additional information

### Funding

| Funder | Grant reference number | Author |
|---|---|---|
| Ministero dell'Istruzione, dell'Università e della Ricerca | RBFR1332DJ | Roberto Arrighi |
| European Research Council | FP7-IDEAS-ERC 338866 | David Charles Burr |
| Ministero della Salute | GR-2013-02358262 | Giovanni Anobile |

The funders had no role in study design, data collection and interpretation, or the decision to submit the work for publication.

### Author contributions

GA, Wrote the paper, Analysis and interpretation of data, Performed research, Designed the research, Conception and design, Acquisition of data ; RA, Wrote the paper, Performed research, Designed the research, Conception and design, Acquisition of data, Analysis and interpretation of data; IT, Performed research, Designed the research, Acquisition of data, Analysis and interpretation of data; DCB, Wrote the paper, Designed the research, Conception and design, Analysis and interpretation of data

## Author ORCIDs

Giovanni Anobile, http://orcid.org/0000-0003-2796-0661
Roberto Arrighi, http://orcid.org/0000-0002-5435-6729
David Charles Burr, http://orcid.org/0000-0003-1541-8832

## Ethics

Human subjects: All participants gave written informed consent. Experimental procedures were approved by the local ethics committee [Comitato Etico Pediatrico Regionale-Azienda Ospedaliero-Universitaria Meyer-Firenze (FI)] and are in line with the declaration of Helsinki.

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
