## [Decision Letter]

Thank you for submitting your article "A shared numerical representation for action and perception" for consideration by eLife. Your article has been favorably evaluated by Timothy Behrens (Senior editor) and three reviewers, one of whom, Richard Ivry (Reviewer #1), is a member of our Board of Reviewing Editors. The following individuals involved in review of your submission have agreed to reveal their identity: Stanislas Dehaene (Reviewer #2); Daniel Casasanto (Reviewer #3).

The reviewers have discussed the reviews with one another and the Reviewing Editor has drafted this decision to help you prepare a revised submission

Summary:

The reviewers found considerable merit in this paper. While many studies have examined number perception, very few have studied number production, and the finding of a shared level of representation (and a new numerical illusion!) is novel. The experiments are well conducted and the results appear to be quite robust. There are two major issues for revision.

Essential revisions:

1) A key claim of the paper is that the effect is perceptual in nature and not cognitive. This conclusion rests of the observation that the action-perception interaction is observed when the number displays are presented in the right visual field and the right hand is used for the production task, but now when the number displays are presented in the left visual field. Thus, the "adaptation" effects appear to be location specific. This is a surprising result given that many studies in the perceptual domain suggest that numeric representation is relatively abstract. However, the current design confounds adapted/unadapted side with right/left hand (or side of space). We would like to have additional experimental work to unconfound these factors. This work will also offer an opportunity to obtain a replication of the location specificity of the effect. Here are a few of the experimental manipulations the authors should consider to further develop the argument that the effect is perceptual and not "cognitive".

A) Repeat the basic experiment but have half the participants tap with left hand and half tap with right hand. This is an essential experiment to run; note that we want to see the right hand effect repeated rather than just have the left hand condition added to the current data set.

B) A second experimental question would be to unconfound hand and space. That is, the right hand condition could be repeated with the person tapping on the left side of the screen. Note that this issue could be addressed in combination with the left vs. right hand experiment described above. The authors need not run a full-factorial design here (fully cross hand and side of movement); a partial design would be fine as long as it unconfounds adapted/unadapted and right/left, with some subset of conditions to unconfound hand and space.

C) The vision-only condition provides an opportunity to further demonstration the location specificity of this adaptation effect. In the current vision-only condition, the adapter and test stimuli are always on the same side of space. It would have been nice to have trials in which they are on opposite sides of space (which might present issues related to shifts of attention, or monitoring eye movements to ensure participants maintain fixation at the center). If this has been done in previous work, please cite this work. If not, it would be a nice additional condition, but we will not require these data in the revision.

2) As noted above, the claim that this effect is perceptual, not cognitive is one of the most intriguing findings in this work. The reviewers were surprised that this finding was noted in the Results section, but not included in the Discussion. Rather, there is discussion of number as a "high-generalized system, capable of integrating numerical information from different senses, and across formats". How do we reconcile this statement with the location specificity and claim of a perceptual basis for the effect? More generally, the conclusion about modality (or location) specificity seems at odds with the large literature showing that number representation is "abstract" and "amodal", work done in people (babies and adults), as well as in other primates. Much of the evidence comes from cross-modal paradigms. This issue should be addressed in the revision, with the authors working to provide an integration of the literature that has focused on number representation being modality specific (work cited in the current version) and the literature pointing to a more abstract representation. We can imagine that both views may be valid, with interactions arising at multiple levels of processing. We do want the authors to provide a better connection between these literatures. Some papers to consider that have advanced the amodal perspective.

Jordan, K. E., Brannon, E. M., Logothetis, N. K., & Ghazanfar, A. A. (2005). Monkeys match the number of voices they hear to the number of faces they see. Current Biology, 15(11), 1034-1038.

Jordan, K. E., & Brannon, E. M. (2006). The multisensory representation of number in infancy. Proceedings of the National Academy of Sciences of the United States of America, 103(9), 3486-3489.

Brannon, E. M. (2003). Number knows no bounds. Trends in cognitive sciences, 7(7), 279-281.

Izard, V., Sann, C., Spelke, E. S., & Streri, A. (2009). Newborn infants perceive abstract numbers. Proceedings of the National Academy of Sciences, 106(25), 10382-10385.

Barth, H., Kanwisher, N., & Spelke, E. (2003). The construction of large number representations in adults. Cognition, 86(3), 201-221.

Barth, H., La Mont, K., Lipton, J., & Spelke, E. S. (2005). Abstract number and arithmetic in preschool children. Proceedings of the National Academy of Sciences of the United States of America, 102(39), 14116-14121.

---

## [Author Response]

*Essential revisions:*

*1) A key claim of the paper is that the effect is perceptual in nature and not cognitive. This conclusion rests of the observation that the action-perception interaction is observed when the number displays are presented in the right visual field and the right hand is used for the production task, but now when the number displays are presented in the left visual field. Thus, the "adaptation" effects appear to be location specific. This is a surprising result given that many studies in the perceptual domain suggest that numeric representation is relatively abstract. However, the current design confounds adapted/unadapted side with right/left hand (or side of space). We would like to have additional experimental work to unconfound these factors. This work will also offer an opportunity to obtain a replication of the location specificity of the effect. Here are a few of the experimental manipulations the authors should consider to further develop the argument that the effect is perceptual and not "cognitive".*

A) Repeat the basic experiment but have half the participants tap with left hand and half tap with right hand. This is an essential experiment to run; note that we want to see the right hand effect repeated rather than just have the left hand condition added to the current data set.

We have done this, not with half the subjects but with a repeated design on all new subjects. The effect was slightly reduced with the non-preferred hand, but definitely confirmed on both hands.

B) A second experimental question would be to unconfound hand and space. That is, the right hand condition could be repeated with the person tapping on the left side of the screen. Note that this issue could be addressed in combination with the left vs. right hand experiment described above. The authors need not run a full-factorial design here (fully cross hand and side of movement); a partial design would be fine as long as it unconfounds adapted/unadapted and right/left, with some subset of conditions to unconfound hand and space.

We have now done this, with very clear results (please see Figure 2). We examined three of the four conditions (omitting the fourth as the effect with the non-preferred hand was weaker than with the preferred). We used the most revealing condition, with simultaneous presentation of a dot area with minimal feedback: motor to sensory, time to space.

C) The vision-only condition provides an opportunity to further demonstration the location specificity of this adaptation effect. In the current vision-only condition, the adapter and test stimuli are always on the same side of space. It would have been nice to have trials in which they are on opposite sides of space (which might present issues related to shifts of attention, or monitoring eye movements to ensure participants maintain fixation at the center). If this has been done in previous work, please cite this work. If not, it would be a nice additional condition, but we will not require these data in the revision.

Indeed, this was a condition that we explored thoroughly in our Proc. Roy. Soc.paper, examining not only spatial selectivity, but also its invariance with eye-movements. We now cite this more directly (Discussion, third paragraph), and make the comparison with the allocentric spatial selectivity revealed here.

*2) As noted above, the claim that this effect is perceptual, not cognitive is one of the most intriguing findings in this work. The reviewers were surprised that this finding was noted in the Results section, but not included in the Discussion. Rather, there is discussion of number as a "high-generalized system, capable of integrating numerical information from different senses, and across formats". How do we reconcile this statement with the location specificity and claim of a perceptual basis for the effect? More generally, the conclusion about modality (or location) specificity seems at odds with the large literature showing that number representation is "abstract" and "amodal", work done in people (babies and adults), as well as in other primates. Much of the evidence comes from cross-modal paradigms. This issue should be addressed in the revision, with the authors working to provide an integration of the literature that has focused on number representation being modality specific (work cited in the current version) and the literature pointing to a more abstract representation. We can imagine that both views may be valid, with interactions arising at multiple levels of processing. We do want the authors to provide a better connection between these literatures. Some papers to consider that have advanced the amodal perspective.*

*Jordan, K. E., Brannon, E. M., Logothetis, N. K., & Ghazanfar, A. A. (2005). Monkeys match the number of voices they hear to the number of faces they see. Current Biology, 15(11), 1034-1038.*

*Jordan, K. E., & Brannon, E. M. (2006). The multisensory representation of number in infancy. Proceedings of the National Academy of Sciences of the United States of America, 103(9), 3486-3489.*

*Brannon, E. M. (2003). Number knows no bounds. Trends in cognitive sciences, 7(7), 279-281.*

*Izard, V., Sann, C., Spelke, E. S., & Streri, A. (2009). Newborn infants perceive abstract numbers. Proceedings of the National Academy of Sciences, 106(25), 10382-10385.*

*Barth, H., Kanwisher, N., & Spelke, E. (2003). The construction of large number representations in adults. Cognition, 86(3), 201-221.*

*Barth, H., La Mont, K., Lipton, J., & Spelke, E. S. (2005). Abstract number and arithmetic in preschool children. Proceedings of the National Academy of Sciences of the United States of America, 102(39), 14116-14121.*

Thank you for this comment. We have now expanded the Introduction to include this work, and mention how our study reinforces and complements the previous work. We agree completely that the number system is abstract and generalized over modalities, formats and even action. However, it is strangely selective spatially. We do not think this in anyway contradicts the generality of the number system, and now discuss this more, making analogies with perception of event duration (also spatially specific). We agree that this is an important point, thank you for pointing it out.